# A Concept Analysis of Maternal Resilience against Pregnancy-Related Mental Health Challenges in Low- and Middle-Income Countries

**DOI:** 10.3390/healthcare12161555

**Published:** 2024-08-06

**Authors:** Anila Naz AliSher, Samia Atta, Adnan Yaqoob, Tanseer Ahmed, Salima Meherali

**Affiliations:** 1Faculty of Nursing, University of Alberta, Edmonton, AB T6G 2R3, Canada; meherali@ualberta.ca; 2College of Nursing, University of Health Sciences, Dera Ghazi Khan 03222, Pakistan; samia.atta47@gmail.com; 3Shaukat Khanum Memorial Cancer Hospital and Research Centre, Lahore 54780, Pakistan; adnanyaqoob@skm.org.pk; 4Division of Nursing, Midwifery, and Social Work, The University of Manchester, Manchester M13 9PL, UK; tanseer.ahmed@postgrad.manchester.ac.uk

**Keywords:** concept analysis, maternal resilience, pregnancy outcome, childbirth

## Abstract

Suicide accounts for 33% of deaths of women during the postnatal period in many low- and middle-income countries (LMICs). Resilience refers to an ability to adapt and recover from adversity or misfortune. Resilience building against mental health challenges during pregnancy and the postnatal period is critical for women to raise their child efficiently and maintain a healthy life. The exploration of maternal resilience against mental health challenges including its developmental processes and the determinants of its successful or unsuccessful cultivation among mothers during pregnancy and childbirth is of paramount importance. Understanding why a subset of mothers effectively develops resilience while others significantly struggle is critical for devising targeted interventions and support mechanisms aimed at improving maternal well-being. This inquiry not only seeks to delineate the factors that contribute to or hinder the development of resilience but also aims to inform the creation of comprehensive support systems that can bolster maternal health outcomes. This paper endeavors to present a comprehensive analysis of maternal resilience, aiming to cultivate a nuanced and profound understanding of the concept within the framework of previous traumatic events and adverse pregnancy outcomes in LMICs. The eight-step method approach proposed by Walker and Avant was utilized for this concept analysis. Several defining attributes were identified in the analysis including social adaptation, support system, optimistic approach, and mindfulness. This analysis contributes to knowledge advancement regarding maternal resilience and provides nurses and other healthcare professionals with a clear understanding of the concept of maternal resilience to help promote resilience among mothers.

## 1. Introduction

Globally, approximately 10% of women during pregnancy and 13% during postpartum encounter mental health issues, predominantly depression. The prevalence rates are even higher in low- and middle-income countries (LMICs), reaching 15.6% during pregnancy and 19.8% postpartum [1]. In extreme instances, maternal distress may escalate to the point of suicidal ideation or action. Research has revealed that suicide accounted for 33% of deaths in the postnatal period in Mozambique, 16% in Nepal, and 13% in Sri Lanka [2,3].

Problems related to maternal and child health are mainly preventable but still account for over 50% of the total disease burden [4]. Providing standard maternal and child health in LMICs is still a significant challenge in the healthcare sector [5]. One of the greatest challenges in obstetric healthcare in the community and hospitals in LMICs is to guarantee a positive experience during prenatal care, assuring health promotion with social, cultural, emotional, and psychological aspects [6,7]. A woman’s experiences during the pre-and postnatal period largely frame her psychological well-being and influence her ability to navigate the challenges of motherhood. Reports from LMICs have revealed a lack of access to health care services, unhygienic practices by birth attendants, substandard care, and lack of resources for equipment and medication are factors contributing to maternal and neonatal deaths and obstetrical complications [8,9]. According to a recent report, almost 2.4 million neonatal deaths have been reported in the past few years [10], and is one of the biggest reasons for the development of postpartum depression among women in LMICs [11].

Becoming a mother is a stressful transition in a woman’s life involving long-term processes reorganizing internal thoughts and external behaviors [12]. This transition can be fraught with difficulties and added stress, making mothers more prone to postpartum depression [13]. A significant proportion of women encounter varying degrees of postpartum depression and anxiety disorders, with the situation exacerbating in instances of neonatal loss [12,13]. Moreover, numerous other stressful events during pregnancy, for example, domestic violence and inability to afford the cost of care, have a significant impact on a woman’s physical, mental, and emotional health and behaviors, which eventually leads to an increased risk of psychological trauma [14]. On top of that, previous traumatic experiences such as postpartum hemorrhage, infection, deep vein thrombosis, pregnancy-related anxiety, and psychosis can lead many women to permanently hesitate about conceiving another child at any point in their lives [15]. Maternal anxiousness further leads to adverse pregnancy outcomes such as preterm birth or complications and can jeopardize their overall mental health.

Resilience is the capacity to recover from stress or adversity. It plays a pivotal role in helping mothers navigate the physical and emotional challenges associated with pregnancy and childbirth [16]. Studies have shown that resilient mothers are better equipped to handle the stressors of pregnancy such as hormonal changes, physical discomfort, and the anxiety of impending motherhood [2,4,9,16]. Resilience building against mental health challenges during pregnancy and the postnatal period is critical for women to raise their child efficiently and maintain a healthy life [4,15]. Furthermore, resilience has been linked to lower levels of postpartum depression and anxiety, contributing to better overall mental health outcomes for both the mother and the infant [17,18]. There is a need to explore the process of developing resilience in order to facilitate a mother’s capacity to navigate mental health challenges in the context of LMICs.

Maternal resilience varies between low- and middle-income countries (LMICs) and high-income countries due to different socioeconomic and cultural contexts [9,19]. In high-income countries, maternal resilience is often supported by greater access to healthcare services, social welfare programs, and educational resources [18,19]. Conversely, in LMICs, maternal resilience is shaped by factors such as limited access to healthcare, economic instability, and higher rates of social adversity [20]. Women in these regions might rely more heavily on community support, traditional practices, and personal resourcefulness to navigate challenges [20,21].

Past studies have explored various facets of maternal resilience including the impact of social support networks, economic stability, and access to healthcare on maternal well-being [4,5,7,9,18,22]. Research has also examined the role of cultural practices and community resources in fostering resilience among mothers in LMICs [7,9,10,12,15,23]. While it is true that maternal mental health and resilience have begun to gain some attention, the issue in the context of LMICs has received comparatively limited attention, which has made it challenging to fully understand and effectively support maternal resilience in LMIC contexts. Given the unique challenges faced by mothers in LMICs such as limited healthcare resources, socioeconomic stressors, and cultural stigmas associated with mental health, fostering resilience is particularly crucial [18,19,24]. Studies indicate that providing expectant mothers with robust support networks and resources such as community-based prenatal education programs, mental health counselling, and peer support groups can significantly enhance their resilience [25,26]. These initiatives help mitigate the impact of stress and adversity, allowing mothers to better cope with the demands of pregnancy and childbirth [27]. Promoting positive coping mechanisms such as mindfulness practices and stress management techniques can further strengthen maternal resilience in LMICs [28]. The benefits of such interventions are profound, extending beyond the individual mother to positively impact infant development and overall family dynamics [29].

Therefore, there is a dire need to explore the process of developing resilience and understand why only a few mothers in LMICs successfully build resilience while many others fail to foster adequate resilience during pregnancy and childbirth. A comprehensive concept analysis of maternal resilience in the context of LMICs is crucial to clarify its definition, identify key components, and guide research, practice, and policy development.

Concept analysis is a strategy used to further develop concepts; it has been presented in the nursing literature as the central part of theory development [18,19,22,24]. It provides a clear definition of the concepts in conjunction with their uses, defining attributes, related concepts, and their applicability to the selected discipline [16,18,22]. Such insights are crucial for nurses as well as other healthcare providers to offer counselling, education, and care services to mothers facing any crisis related to childbirth as well as to assist them with challenges associated with pregnancy outcomes and low socioeconomic status [25,26,28].

This paper aimed to dissect the construct of maternal resilience in the context of LMICs within the purview of previous traumatic events encountered during pregnancy and the postnatal period. Utilizing Walker and Avant’s concept analysis methodology [30], this study delineated the defining attributes, empirical referents, antecedents, and consequences of maternal resilience. Furthermore, a secondary objective was to establish an operational definition of maternal resilience, thereby contributing to a deeper understanding and more effective support mechanisms for mothers facing such challenges in LMICs.

## 2. Materials and Methods

Various search strategies were employed to identify relevant literature on the topic. The necessary literature was retrieved from multiple databases including PubMed, Medline, the Cumulative Index to Nursing and Allied Health Literature (CINAHL) Plus with Full Text, and PsycINFO. Additionally, relevant reports concerning mothers from low-resource settings were sourced from websites such as the World Health Organization (WHO) and the United Nations International Children’s Emergency Fund (UNICEF). The keywords used in the search included ‘maternal’, ‘resilience’, ‘pregnancy’, ‘postpartum’, ‘perinatal’, ‘antenatal’, ‘prenatal’, ‘coping’, ‘psychological’, ‘postnatal’, ‘expectant’, ‘motherhood’, ‘traumatic’, ‘trauma’, ‘adverse’, ‘stressful’, ‘poor outcomes’, ‘pregnancy complications’, ‘adverse pregnancy outcomes’, and ‘birth outcomes’. A manual search was also conducted using references from the relevant articles. The search was limited to English-language and full-text articles published between 1990 and 2024.

The data analysis for the concept analysis of maternal resilience was conducted by using Walker and Avant’s eight-step method to identify conceptual components of maternal resilience [30]. The steps included: (1) choose the concept, (2) determine the reason for analysis, (3) identify all uses of the concept, (4) identify key attributes that make up the concept, (5) develop a model case, (6) construct additional cases that do not reflect the concept, (7) identify pre-conditions and outcomes, and (8) determine empirical referents. This analysis aimed to clarify the definition of maternal resilience, identify its attributes, antecedents, and consequences, and develop a comprehensive understanding of how it impacts mothers during and after pregnancy in the context of LMICs.

The authors involved in the data analysis process possessed substantial expertise, ensuring a rigorous approach to the study. One author has a robust background in qualitative inquiry and has published numerous qualitative research papers. Another author specializes in maternal and child health and is currently pursuing a PhD with a research focus on maternal health. The third author, who supervised the entire process, is a senior researcher with extensive experience in this area and has published several qualitative and quantitative research papers. Their combined expertise contributed to a meticulous and thorough analysis of the data.

## 3. Results

A total of 3736 citations were retrieved from various databases, with an additional 13 citations sourced from references, bringing the initial total to 3749 (Figure 1). After removing 1717 duplicate references, 2019 studies remained for screening. Title and abstract screening excluded 1739 citations, leaving 280 articles for full-text retrieval. Out of these, 277 studies were assessed for eligibility. Subsequently, 268 studies were excluded, with 267 not related to maternal resilience and 15 not conducted in LMICs. This process resulted in seven studies being included for data extraction and analysis. The data extracted from these studies are intended to provide comprehensive insights into maternal resilience during and after pregnancy.

The analysis revealed that the included studies utilized the term ‘maternal resilience’. However, none of these studies offered a clear definition of the term in the context of LMICs. It seems that the authors employed ‘maternal resilience’ to link the idea of resilience specifically to the perinatal period and motherhood rather than identifying unique qualities or aspects of resilience relevant to this life stage being located in a low- and middle-income country. This highlights the necessity for a more precise and consistent definition of maternal resilience in future research to adequately reflect its distinctive characteristics during and after pregnancy. Table 1 provides the details of characteristics of included studies. 

### 3.1. Overview of the Use of Concept

Resilience is used as a buzzword in different professions and is a widely used concept in almost all disciplines including medicine, psychology, social sciences, microbiology, ecology, and nursing. It is an ability to recover from or adjust easily to misfortune or change [36]. Resilience is a person’s aptitude for adapting, recovering, and functioning during times of adversity [18]. Gillespie et al. argue that resilience is used to support the idea of adapting well to adversity and bouncing back from difficult experiences [15]. Moreover, resilience is defined by psychologists as the process of adaptation in times of adversity, trauma, tragedy, threats, or significant sources of stress such as relationship problems, serious health problems, or financial stressors [20]. Furthermore, the factors that affect an individual’s positivity in maintaining a healthy lifestyle despite facing trauma and misfortune in life are collectively called resilience [21,22].

In nursing, the concept of resilience has been studied by several scholars in different contexts such as in chronic pain [21], academic resilience [22], caregiver’s resilience [23], perinatal resilience [24], and family resilience [25]. Though the concept of resilience is not unique in nursing, it is noted that there are scanty resources available in the nursing literature regarding maternal resilience in low socioeconomic contexts. Maternal resilience developed among the mothers who bounced back from negative experiences and turbulences in life while pregnant [22]. Verner et al. defined maternal resilience as the extent to which a mother is capable of maintaining positivity and social interactions during pregnancy and childbirth [24]. Pregnancy is considered a period of emotional variations caused by social, psychological, and hormonal changes [5,7,13,25,37].

Some major stressors during pregnancy against which resilience is supposed to be built are unplanned pregnancies, changes in family dynamics such as the relationship with a partner, acquired responsibilities with neonatal care, and, more importantly, the risk of complications during pregnancy and labor [38]. Researchers have identified a few socioeconomic factors of stress including low socioeconomic status, inability to bear the cost of health and childbirth, domestic violence, and lack of a family support network [39]. It is recognized that there is a variation in resilience-building depending on personal characteristics and the context in which women are living and facing different circumstances [37,38]. The factors affecting resilience are addressed differently depending on the situation. Some researchers relate the concept of resilience to social factors such as poverty or trauma [39], while others link it to personal qualities necessary for resilience [40].

To conceptually refine the concept of maternal resilience in the context of LMICs, the literature was re-examined with the help of Walker and Avant’s concept analysis techniques [30,40]. Defining attributes, empirical referents, antecedents, and consequences are shown in Table 2.

### 3.2. Attributes of Maternal Resilience

Attributes are the characteristics of a concept that are linked with the concept very frequently and tend to occur in the literature whenever the concept is being searched [15,16,36]. Based on the literature, social adaptation, support system, optimistic approach and positivity, and mindfulness were identified as attributes of the concept of maternal resilience.

#### 3.2.1. Social Adaptations

This attribute refers to resilience among pregnant women living in low-resource settings and areas where access to healthcare is not easily accessible. Social adaptations refer to their ability to cope with scarce healthcare resources yet exhibit a willingness to stay positive. Their hope of dealing with childbirth-related stress consequently plays a significant role in reducing morbidity and mortality [14]. Despite the lack of healthcare resources, mothers living in difficult situations can still prove to be resilient [27,41], which enables them to thrive in a contented life. Moreover, social adaptation can be influenced by a woman’s personal conditions, cultural and religious beliefs and attitudes, knowledge, community, and societal conditions [42]. Adaptation and resilience can be enhanced by assessment to determine vulnerability, risk, and protective factors [27,41,42]. Nurses and other healthcare providers can assist mothers in assessing their personal conditions, cultural beliefs and attitudes, and community and societal conditions [25]. Once assessed, providers could work collaboratively to assist mothers in developing a plan for reducing risk and increasing protective factors [4,9,23,27,41].

#### 3.2.2. Support System

In the majority of LMICs including regions in South Asia and Africa, it is a norm for women to reside with their husband’s family after marriage [10]. Therefore, family dynamics can impact women in two distinct ways. First, if the relationship with the husband’s family is positive and healthy, it can provide substantial support during pregnancy and the postnatal period [4,5,6,43,44]. Conversely, suppose a woman is unable to establish a harmonious relationship with her in-laws, which is often the case. In that case, she may find herself devoid of support not only from her in-laws, but also from her spouse and her own family [5,6,13]. It is frequently reported that an unstable relationship with an intimate partner, encompassing experiences of spousal violence and disputes with in-laws, has been identified as a risk factor for mental health issues in the postnatal period among women in LMICs [45].

In some private hospitals of LMICs, prenatal and postnatal educational sessions are being offered to expectant mothers, which is characterized by high healthcare costs [45,46]. Though these sessions have demonstrated considerable efficacy in providing support to new mothers, accessibility to such sessions remains a challenge for many women due to its cost [5,45]. Additionally, compared to educational sessions, community health workers and midwives in many rural areas of LMICs frequently visit pregnant women and provide substantial support in terms of health monitoring, nutritional advice, and emotional counselling, all free of cost or at a minimum cost [9,11], which represents a significant benefit for pregnant women in terms of support. This level of support can differ according to social, religious, and cultural backgrounds [3,4,9,12]. Mothers can seek social support from other pregnant women in the same context as well as from healthcare professionals including nurses, gynecologists and family members, especially sisters and cousins who have experience as mothers [42].

Researchers have identified the role of technology use in social support among pregnant women by promoting their feelings of being integrated into a large social network [43]. Effective social support can make mothers feel valued and cared for and provide a sense of belonging [44]. Therefore, it is considered as a strong attribute of maternal resilience. Based on the data from previous studies, support systems are considered an important attribute of maternal resilience in the postnatal period [3,4,9,12,44]. This support system can effectively reduce the risk of depression during pregnancy for mothers as well as lead them toward good health and pregnancy outcomes [46].

#### 3.2.3. Optimistic Approach

The terms positivity and optimism are sometimes used interchangeably and are considered an important attribute of resilience [18,45,46]. They refer to experiencing positive emotions and directing life’s challenges and hardships in a positive way [47]. Kułak-Bejda and colleagues define optimism as a mindset reflecting a belief that the outcome of a particular endeavor will be desirable [48]. It was found that women with higher optimism during pregnancy have less stress, anxiety, and pre- and postpartum depression than women with lower optimism [49]. Optimism has also been linked to birth outcomes, with one study showing that optimistic women gave birth to healthier babies [50]. Optimism in a person is shaped by biological elements, especially heredity and temperament as well as environmental elements including parental influence, guidance from teachers, and early life experiences [48]. Though optimism is an intrinsic constituent of an individual’s personality, this is an acquired skill, not an inherent trait, and any woman can develop it with the help of perseverance, positive reinforcement, and supportive communities [51].

In previous pregnancies, women who experienced adverse outcomes, especially stillbirth and neonatal death, may exhibit reduced levels of optimism in subsequent pregnancies. Nevertheless, healthcare providers, along with support from family, peers, and spouses, can exert a considerable influence in fostering optimism among pregnant women.

#### 3.2.4. Mindfulness

Pregnancy is a time of major changes in a woman’s life; increased stress as well as mood and sleep disorders are common. Psychological changes in a pregnant woman who assumes a maternal role are crucial in the construction of maternal identity by redefining a woman’s self-perception and the physical and emotional modifications in her sociocultural dynamics [22]. The steadfast pursuit of objectives and engagements in the face of adversities and adaptive goal recalibration underscores the paramount significance of the maternal resilience-building process. Active coping includes pursuing activities or goals as much as possible within the parameters of problems and engaging in techniques that reduce hopelessness like sharing and catharsis or mindfulness meditation [52]. Mindfulness during pregnancy boosts a mother’s ability to manage stress, anxiety, depression, emotional disturbances, and sleep patterns [53].

At one end of the spectrum lies resilience, while at the opposite end are mental health issues. Tilting the balance toward resilience among mothers requires integrating four intertwined concepts: support systems, social adaptation, optimism, and mindfulness. These elements are intricately linked with each other. Thus, disruption in one may impact the others [54].

### 3.3. Antecedents and Consequences

Antecedents and consequences in a concept analysis are often overlooked or taken for granted, but they play pivotal roles in elucidating the full scope and implications of the concept under examination.

#### 3.3.1. Antecedents

Antecedents are those events that must occur before the occurrence of an incident [16]. A few significant antecedents of maternal resilience are hardships and troubles, unaddressed health necessities [8], lack of access to resources, and past traumatic events related to childbirth, which give rise to the fear of impending doom during pregnancy [54], helplessness, and a lack of family support [12]. Many women face social isolation and abandonment during pregnancy, which is as harmful as smoking 15 cigarettes a day (WHO, 2024). Moreover, psychological distress [54], neonatal complications including low birth weight [21], stress [28], powerlessness [55], vulnerability [56], and inferior psychosocial outcomes [50] are also a few examples of antecedents of maternal resilience.

#### 3.3.2. Consequences

A consequence is a result or outcome that follows as a natural effect or inference from an action, event, or condition. It signifies the implications, effects, or repercussions that arise as a direct or indirect result of a particular cause or situation. Walker and Avant delineated consequences as the outcomes of an event [30]. If a woman successfully fosters resilience after navigating antecedents, it will result in gaining confidence, motivation, and improvements in health [50,54,55]. However, if she fails, she may end up experiencing diminished self-efficacy, potential deterioration in mental health, and possibly adverse effects on her overall well-being [3,48]. In Figure 2, the interplay of attributes of resilience is exhibited.

### 3.4. Empirical Referents

The last step of concept analysis is the identification of the referents of the concept, which demonstrates the existence of the concept and explores how the concept and its attributes have been measured [16]. Empirical referents serve as tangible markers or indicators that help researchers understand, measure, and apply abstract concepts in practical contexts [30]. They bridge the gap between theoretical conceptualizations and empirical observations, facilitating the systematic analysis and evaluation of complex phenomena. In the nursing discipline, no instrument to measure maternal resilience has been developed. However, there are several examples of resilience instruments designed to measure it in other contexts. In the context of racial/ethnic differences in low birth weight, Montoya-Williams et al. utilized the 10-item Connor–Davidson Resilience Scale (CD-RISC-10) to measure resilience among pregnant women [19]. Campbell-Sills et al. validated the tool [50], and it was applied to many different groups, especially community samples, primary care outpatients, general psychiatric outpatients, clinical trials of generalized anxiety disorder, and two clinical trials of PTSD, cancer patients, and diabetic patients [19,50].

### 3.5. Operational Definition

Through an examination of defining attributes, antecedents, and consequences, we aimed to formulate an operational definition of maternal resilience. Maternal resilience can be construed as the capacity of mothers to persevere with a sense of optimism, striving for a fulfilling existence while nurturing their newborn, which entails overcoming various stressors, encompassing concerns such as the fear of infant loss, birth complications, and challenges encountered during pregnancy and the postnatal period. These challenges may include unmet health needs, limited access to resources, feelings of helplessness, social isolation, abandonment, and a lack of familial and spousal support.

## 4. Discussion

The concept of maternal resilience for mothers is multifaceted and complex. Based on the concept analysis, we tend to define maternal resilience as the capacity of mothers to persevere with a sense of optimism, striving for a fulfilling existence while nurturing their newborn.

Results of this paper exhibited that maternal resilience in the LMIC context is influenced by multiple factors including hardships and troubles during the pregnancy and postnatal period, unaddressed health necessities, lack of access to resources, past traumatic events related to childbirth, helplessness, and social isolation/abandonment [50,54]. Coping with these factors effectively requires specific characteristics including social adaptation, a support system, optimism, and mindfulness. These take women toward confidence in themselves, motivation toward life, and overall health improvements [45,46].

The accessibility of adequate healthcare services for pregnant women in low socioeconomic areas has emerged as a critical factor in mitigating stress and promoting maternal resilience. The findings of this study underscore the role of familial support, peer networks, and the availability of responsive healthcare providers as pivotal elements in assisting these women to thrive amidst adversity. However, the data revealed a concerning trend: a subset of these women lacks such support systems, which exacerbates mental health issues within their already stressed psyches [3,5,6,7,10,29,56]. This absence of support undermines their confidence and positivity in life [26,29,50]. In contrast, a robust support network has been consistently identified in the literature as instrumental in fostering maternal well-being [2,13].

Women who experience neonatal death are particularly susceptible to postpartum depression, necessitating more comprehensive treatment approaches to facilitate their return to a stable life [2,9,19]. Family and spousal support are significantly impactful in managing such traumatic experiences effectively. Moreover, low birth weight has been identified as a significant marker of a woman’s vulnerability to mental health issues, suggesting that interventions from healthcare providers and government initiatives to ensure proper nutrition for pregnant women in LMICs could enhance resilience to these stressors [9,19,45,48,50,54]. Assurance regarding the health of their baby enables women to build resilience against other adversities [3,6,9].

The role of the woman’s immediate social environment including in-laws and spouses is also of paramount importance [31,47,52,53]. Education and counselling for these family members about the hormonal changes and challenges faced by pregnant women can foster an environment of empathy and support [45,56]. Such understanding could encourage more practical assistance with household tasks and provide the moral support that is often lacking in many South-Asian LMICs [5,6,7,9,32,53].

Additionally, hospitals and community centers in LMICs frequently face issues of overcrowding and understaffing, leading to healthcare providers experiencing burnout [5,9,48]. This, in turn, can result in a diminished capacity for empathetic and attentive patient care. It is imperative that policies be implemented to manage the high patient-to-staff ratio effectively [28,32,33,39].

Finally, the development of maternal resilience is not solely dependent on the woman; it is inextricably linked to a multitude of internal and external antecedents [14,18,44]. Therefore, comprehensive interventions must address all of these dimensions, which is only feasible through collaborative efforts encompassing the pregnant woman, her environmental context, and the healthcare infrastructure of LMICs [19,35,39].

Enhancing adaptation and resilience involves conducting assessments to determine vulnerability, risk, and protective factors. Nurses and other healthcare providers can assist mothers in assessing their personal conditions, cultural beliefs and attitudes, and community and societal conditions [25,34,44,45,48]. Once assessed, providers could work collaboratively to assist mothers in developing a plan for reducing risk and increasing protective factors [5,39,49].

### Implications

Understanding the factors that contribute to resilience enables the development of personalized healthcare plans. Moreover, healthcare providers can tailor interventions to bolster resilience among mothers, particularly those identified as at risk of developing severe mental health challenges during their pregnancy and postnatal periods. The early identification of mothers who may struggle with developing resilience can lead to timely and targeted interventions, potentially mitigating mental health challenges [18,27,54].

The insights from this analysis can inform health policy, advocating for the inclusion of resilience-building programs in maternal healthcare policies, especially in LMICs where resources are scarce. Health programs can be designed to incorporate resilience training and support as core components, ensuring that maternal mental health is prioritized and addressed comprehensively. Educating healthcare professionals about the concept of maternal resilience and its defining attributes can enhance their ability to support and promote resilience among mothers [33,50,53].

Future research should consider longitudinal studies on maternal resilience, exploring cultural contexts and broader determinants like socioeconomic factors and healthcare access [18,21,49]. Addressing the high rates of maternal suicide in LMICs by adapting resilience strategies to these settings is vital. Integrating these findings into practice, research and policy can better support first-time mothers, improving maternal and child health globally.

## 5. Conclusions

Due to diverse and multiple uses of maternal resilience in different contexts, the concept was not well-developed and therefore lacks a specific definition in terms of its use for mothers who had previous traumatic events and are facing poor pregnancy outcomes. Using Walker and Avant’s method [16,30], we constructed a definition by keeping in mind all of the theoretical, conceptual, and philosophical aspects of the concept. Nurses at the hospital as well as at the community level often encounter situations like postpartum depression, anxiety, and stress among pregnant women. We anticipate that improving the nurses’ understanding of the concept of maternal resilience can help in the early detection of impaired coping and refine nursing interventions to promote resilience among them. Moreover, this helps nurses to be thorough and systematic in their efforts to improve their clients’ health outcomes and can develop knowledge for future theory, practice, and research.

## Figures and Tables

**Figure 1 healthcare-12-01555-f001:**
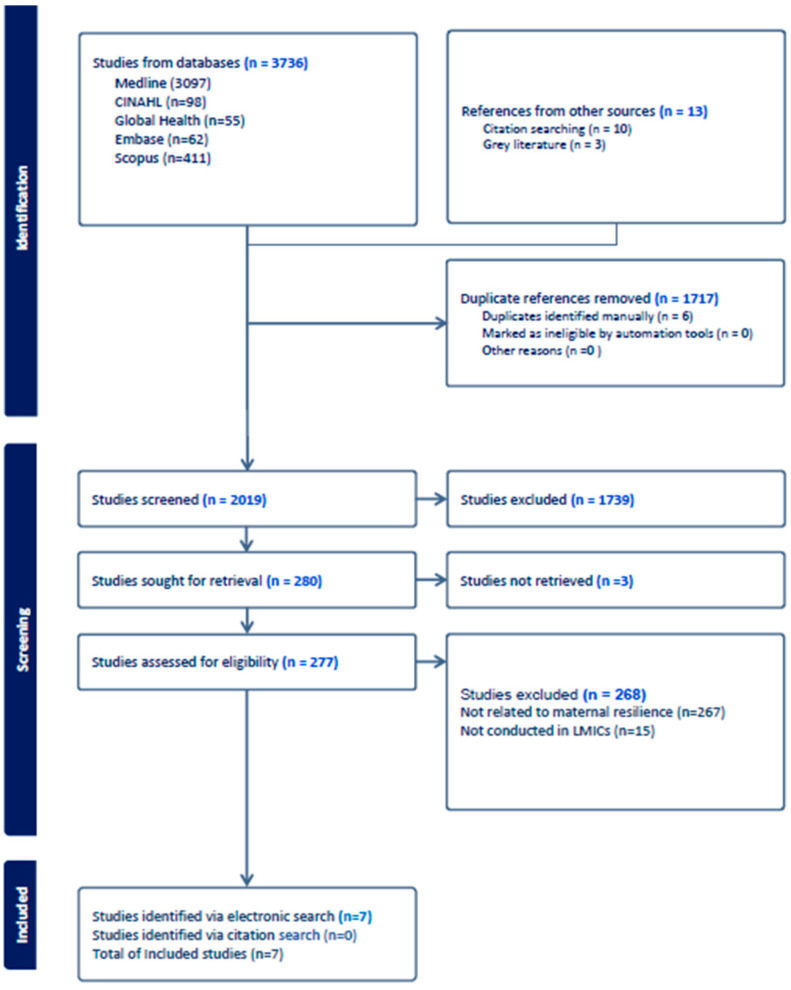
PRISMA flow diagram.

**Figure 2 healthcare-12-01555-f002:**
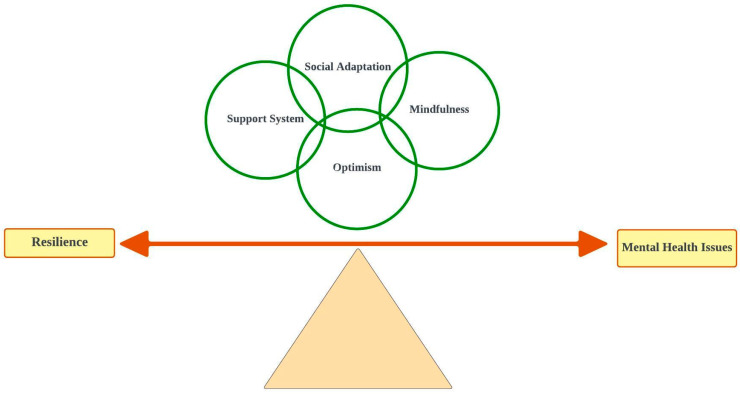
The interplay of attributes of resilience.

**Table 1 healthcare-12-01555-t001:** Summary of the study characteristics.

Authors and Year	Country	Study Design	Study Objectives	Population and Sample	Titles
Dadi et al., 2020 [10]	Ethiopia	Community-based cohort study.	To identify the incidence and predictors of adverse birth outcomes, focusing on the outcomes resulting from exposure to antenatal depression and other psychosocial risk factors during pregnancy in Gondar Town, Ethiopia.	916 pregnant women.	Effect of antenatal depression on adverse birth outcomes in Gondar town, Ethiopia: A community-based cohort study.
Dadi et al., 2020 [31]	Ethiopia	Community-based cohort study.	Aim to explore the causal mechanisms underlying antenatal depression in Gondar, Ethiopia.	916 pregnant women.	Antenatal depression and its potential causal mechanisms among pregnant mothers in Gondar town: Application of structural equation model.
Shawon et al., 2024 [32]	Nepal	Quantitative cross-sectional.	This study investigates the associations between women empowerment and the prevalence of mental health symptoms and care-seeking behavior among ever-married Nepalese women aged 15–49 years.	5556 married women.	Role of women empowerment on mental health problems and care-seeking behavior among married women in Nepal: secondary analysis of nationally representative data.
Nabwera et al., 2018 [33]	Gambia	Case–control study.	To explore the influence of maternal psychosocial circumstances and physical environment on the risk of severe wasting in rural Gambia.	97 cases and 291 controls.	The influence of maternal psychosocial circumstances and physical environment on the risk of severe wasting in rural Gambian infants: a mixed methods approach.
Bhamani et al., 2023 [34]	Pakistan	Multi-phase interventional research	Development and validation of the Safe Motherhood-Accessible Resilience Training (SM-ART) intervention for pregnant women in Pakistan.	17 pregnant women and 8 key informants including psychologists, psychiatrists, and nurses.	Development and Validation of Safe Motherhood-Accessible Resilience Training (SM-ART) Intervention to Improve Perinatal Mental Health.
Abera et al., 2023 [35]	Ethiopia	Quantitative cross-sectional study	To investigate whether pregnancy is associated with greater stress and lower psychological resilience among women living in Jimma, southwest Ethiopia.	66 pregnant and 154 non-pregnant women.	Stress and resilience during pregnancy: A comparative study between pregnant and non-pregnant women in Ethiopia.
Bhamani et al., 2024 [5]	Pakistan	Single-blinded block randomized controlled study.	To examine the effect of the ‘Safe Motherhood—Accessible Resilience Training (SM-ART)’ on resilience, marital adjustment, depression, and pregnancy-related anxiety.	200 pregnant women.	Promoting mental well-being in pregnant women living in Pakistan with the Safe Motherhood—Accessible Resilience Training (SM-ART) intervention: a randomized controlled trial.

**Table 2 healthcare-12-01555-t002:** Process of maternal resilience.

Item	Characteristics
Antecedents	Hardships and troubles
	Unaddressed health necessities
	Lack of access to resources
	Past traumatic events related to childbirth
	Helplessness
	Social isolation/abandonment
	Lack of family support
Attributes	Social adaptation
	Support system
	Optimism
	Mindfulness
Consequences	Confidence
	Motivation
	Health improvements

## Data Availability

No new data were created or analyzed in this study.

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
