# Peer review of "A Concept Analysis of Maternal Resilience against Pregnancy-Related Mental Health Challenges in Low- and Middle-Income Countries"

_healthcare, 2024, doi:10.3390/healthcare12161555_

Round 1

Reviewer 1 Report

Comments and Suggestions for Authors

In this paper, the authors conducted a concept analysis of maternal resilience in low- and middle-income countries. Maternal resilience is an important concept associated with various health factors for women and therefore vital to explore. However, there are some issues/concerns that need to be addressed.

Introduction

1. Line 37 – insert ‘and’ before ‘13% in’

2. Resilience is barely mentioned until end of introduction. Introduction should be edited to focus less describing mental health issues faced by women in prenatal and post-natal periods and more about resilience.

3. It is not enough to say “maternal resilience particular in the context of LMICS has received comparatively limited attention.” What has been done so far and why is there a need for a concept analysis of the term maternal resilience?

4. What is different of maternal resilience between LMICs and high-income countries? Currently, the introduction does not justify a need for the current study.

5. Look at article by Hannon, Daly, & Higgins (2022), citation below. While they used a different approach and not LMICs, it would be beneficial to state their findings then go into the need for concept analysis in LMICs.

Hannon, S. E., Daly, D., & Higgins, A. (2022). Resilience in the perinatal period and early motherhood: a principle-based concept analysis. International Journal of Environmental Research and Public Health19(8), 4754.

Methods and Results

6. More information on literature search is needed. How many articles were identified using keyword search, how many with manual search? Were there duplicates or articles excluded, not used? Were any articles used not from low- or middle-income countries as that was not apart of keywords? It seems that the websites, WHO and UNICEF, the only sources of information for low- and middle-income countries as the literature search in databases did not excluded articles are from other countries.

a. A prisma flow chart would be helpful

7. There is very little information about what was done and a supplemental table with a list of all articles used in concept analysis would be beneficial. A table with columns for authors, country, study design (qualitative/quantitative), are needed and either additional columns for antecedents, attributes, and consequences or a single column listing role in concept should be given to see how the concept analysis results were found.

8. 3.2.2 Support System (paragraph 1, lines 190-199)- there are no citations supporting this.

9. Lines 237-239 needs a reference

10. It states previous pregnancies in line 237 but title is “… among first time mothers” Are all articles used for concept only about first-time mothers?  I find it odd the title specifically points at first time mothers but the phrase first-time mothers isn’t used again until the implications (line 383).

Discussion and Conclusion

11. The discussion and conclusion seem a bit wordy and could use editing to make it more concise, especially the implications section.

Comments on the Quality of English Language

Author Response

Introduction

  1. Line 37 – insert ‘and’ before ‘13% in’

Thank you for pointing out. We have inserted “and” before, 13 %.

  1. Resilience is barely mentioned until the end of introduction. Introduction should be edited to focus less describing mental health issues faced by women in prenatal and post-natal periods and more about resilience.

We have now cut short the content related to mental health from our introduction and try to provide a picture of resilience in the context of LMICs:

Resilience is the capacity to recover from stress or adversity, it plays a pivotal role in helping mothers navigate their physical and emotional challenges associated with pregnancy and childbirth [16]. Studies have shown that resilient mothers are better equipped to handle the stressors of pregnancy, such as hormonal changes, physical discomfort, and the anxiety of impending motherhood [2,4,9,16]. Resilience building against mental health challenges during pregnancy and post-natal period is critical for women to raise their children efficiently and maintain a healthy life [4,15]. Furthermore, resilience has been linked to lower levels of postpartum depression and anxiety, contributing to better overall mental health outcomes for both the mother and the infant [17,18]. There is a need to explore the process of developing resilience in order to facilitate a mother’s capacity to navigate mental health challenges in the context of LMICs.

Maternal resilience varies between low- and middle-income countries (LMICs) and high-income countries due to different socio-economic and cultural contexts [9,19]. In high-income countries, maternal resilience is often supported by greater access to healthcare services, social welfare programs, and educational resources [18,19]. Conversely, in LMICs, maternal resilience is shaped by factors such as limited access to healthcare, economic instability, and higher rates of social adversity [20]. Women in these regions might rely more heavily on community support, traditional practices, and personal resourcefulness to navigate challenges [20,21].

Past studies have explored various facets of maternal resilience, including the impact of social support networks, economic stability, and access to healthcare on maternal well-being [4,5,7,918,22,35,43,51]. Research has also examined the role of cultural practices and community resources in fostering resilience among mothers in LMICs [7,9,10,12,15,26, 34, 49]. While it is true that maternal mental health and resilience have begun to get some attention, the issue in the context of LMICs has received comparatively limited attention, which has made it challenging to fully understand and effectively support maternal resilience in LMICs contexts. Given the unique challenges faced by mothers in LMICs, such as limited healthcare resources, socio-economic stressors, and cultural stigmas associated with mental health, fostering resilience is particularly crucial [18,19,24]. Studies indicate that providing expectant mothers with robust support networks and resources, such as community-based prenatal education programs, mental health counselling, and peer support groups, can significantly enhance their resilience [25,26]. These initiatives help mitigate the impact of stress and adversity, allowing mothers to better cope with the demands of pregnancy and childbirth [27]. Additionally, promoting positive coping mechanisms, such as mindfulness practices and stress management techniques, can further strengthen maternal resilience in LMICs [28]. The benefits of such interventions are profound, extending beyond the individual mother to positively impact infant development and overall family dynamics [29].

Therefore, there is a dire need to explore the process of developing resilience and understand why only a few mothers in LMICs successfully build resilience while many others fail to foster adequate resilience during pregnancy and childbirth. A comprehensive concept analysis of maternal resilience in the context of LMICs is crucial to clarify its definition, identify key components, and guide research, practice, and policy development.

  1. It is not enough to say that “maternal resilience, particularly in the context of LMICS, has received comparatively limited attention.” What has been done so far and why is there a need for a concept analysis of the term maternal resilience?

We agree with that. That’s why we have added the following details related to resiliency as you recommended.

Past studies have explored various facets of maternal resilience, including the impact of social support networks, economic stability, and access to healthcare on maternal well-being [4,5,7,918,2235,43,51]. Research has also examined the role of cultural practices and community resources in fostering resilience among mothers in LMICs [7,9,10,12,15,26, 34, 49]. While it is true that maternal mental health and resilience have begun to get some attention, the issue in the context of LMICs has received comparatively limited attention, which has made it challenging to fully understand and effectively support maternal resilience in LMICs contexts. Given the unique challenges faced by mothers in LMICs, such as limited healthcare resources, socio-economic stressors, and cultural stigmas associated with mental health, fostering resilience is particularly crucial [18,19,24]. Studies indicate that providing expectant mothers with robust support networks and resources, such as community-based prenatal education programs, mental health counselling, and peer support groups, can significantly enhance their resilience [25,26]. These initiatives help mitigate the impact of stress and adversity, allowing mothers to better cope with the demands of pregnancy and childbirth [27]. Additionally, promoting positive coping mechanisms, such as mindfulness practices and stress management techniques, can further strengthen maternal resilience in LMICs [28]. The benefits of such interventions are profound, extending beyond the individual mother to positively impact infant development and overall family dynamics [29].

Therefore, there is a dire need to explore the process of developing resilience and understand why only a few mothers in LMICs successfully build resilience while many others fail to foster adequate resilience during pregnancy and childbirth. A comprehensive concept analysis of maternal resilience in the context of LMICs is crucial to clarify its definition, identify key components, and guide research, practice, and policy development.

  1. 4. What is the difference of maternal resilience between LMICs and high-income countries? Currently, the introduction does not justify a need for the current study.

Maternal resilience varies between low- and middle-income countries (LMICs) and high-income countries due to different socio-economic and cultural contexts [9,19]. In high-income countries, maternal resilience is often supported by greater access to healthcare services, social welfare programs, and educational resources [18,19]. Conversely, in LMICs, maternal resilience is shaped by factors such as limited access to healthcare, economic instability, and higher rates of social adversity [20]. Women in these regions might rely more heavily on community support, traditional practices, and personal resourcefulness to navigate challenges [20,21].

  1. Look at article by Hannon, Daly, & Higgins (2022), citation below. While they used a different approach and not LMICs, it would be beneficial to state their findings then go into the need for concept analysis in LMICs.

Hannon, S. E., Daly, D., & Higgins, A. (2022). Resilience in the perinatal period and early motherhood: a principle-based concept analysis. International Journal of Environmental Research and Public Health, 19(8), 4754.

Thank you very much for suggesting such a wonderful and recent work done by Hannon and Higgins, we have now cited this article in our paper.

Methods and Results

  1. More information on literature search is needed. How many articles were identified using keyword search, how many with manual search? Were there duplicates or articles excluded, not used? Were any articles used not from low- or middle-income countries as that was not apart of keywords? It seems that the websites, WHO and UNICEF, the only sources of information for low- and middle-income countries as the literature search in databases did not excluded articles are from other countries.
  2. A prisma flow chart would be helpful

Thanks for this comment, we have addressed this issue and also added a Prisma flow diagram now.

A total of 3736 citations were retrieved from various databases, with an additional 13 citations sourced from references, bringing the initial total to 3749. After removing 1717 duplicate references, 2019 studies remained for screening. Title and abstract screening excluded 1739 citations, leaving 280 articles for full-text retrieval. Out of these, 277 studies were assessed for eligibility. Subsequently, 268 studies were excluded, with 267 not related to maternal resilience and 15 not conducted in low- and middle-income countries (LMICs). This process resulted in 7 studies being included for data extraction and analysis. The data extracted from these studies are intended to provide comprehensive insights into maternal resilience during and after pregnancy.

The analysis revealed that the included studies utilized the term 'maternal resilience.' However, none of these studies offered a clear definition of the term in the context of low- and middle-income countries. It seems that the authors employed 'maternal resilience' to link the idea of resilience specifically to the perinatal period and motherhood rather than identifying unique qualities or aspects of resilience relevant to this life stage being located in a low- and middle-income country. This highlights the necessity for a more precise and consistent definition of 'maternal resilience' in future research to adequately reflect its distinctive characteristics during and after pregnancy.

  1. There is very little information about what was done and a supplemental table with a list of all articles used in concept analysis would be beneficial. A table with columns for authors, country, study design (qualitative/quantitative), are needed and either additional columns for antecedents, attributes, and consequences or a single column listing role in concept should be given to see how the concept analysis results were found.

Thank you, we have included the table now.

Authors

Year

Country

Study Design

Titles

Dadi et al

2020

Ethiopia

Community-based cohort study

Effect of antenatal depression on adverse birth outcomes in Gondar town, Ethiopia: A community-based cohort study.

Dadi et al.

2020

Ethiopia

Community-based cohort study

Antenatal depression and its potential causal mechanisms among pregnant mothers in Gondar town: Application of structural equation model.

Shawon et al

2024

Nepal

Quantitative cross sectional

Role of women empowerment on mental health problems and care-seeking behavior among married women in Nepal: secondary analysis of nationally representative data.

Nabwera et al

2018

Gambia

Case-control study 

The influence of maternal psychosocial circumstances and physical environment on the risk of severe wasting in rural Gambian infants: a mixed methods approach.

Bhamani et al.

2023

Pakistan

Multi-phase Interventional Research

Development and Validation of Safe Motherhood-Accessible Resilience Training (SM-ART) Intervention to Improve Perinatal Mental Health

Abera et al.

2023

Ethiopia

Quantitative cross-sectional study

Stress and resilience during pregnancy: A comparative study between pregnant and non-pregnant women in Ethiopia

Bhamani et al

2024

Pakistan

Single-blinded block randomized controlled study

Promoting mental wellbeing in pregnant women living in Pakistan with the Safe

Motherhood—Accessible Resilience Training (SM‑ART) intervention: a randomized controlled trial

  1. 3.2.2 Support System (paragraph 1, lines 190-199)- there are no citations supporting this.

We have now added the citations.

  1. Lines 237-239 needs a reference

We have added that.

  1. It states previous pregnancies in line 237 but title is “… among first time mothers” Are all articles used for concept only about first-time mothers?  I find it odd the title specifically points at first time mothers but the phrase first-time mothers isn’t used again until the implications (line 383).

We realize this mistake and amend the title and implications as well.

A concept Analysis of maternal resilience against pregnancy-related mental health challenges in LMICs

4.1. Implications

Understanding the factors that contribute to resilience enables the development of personalized healthcare plans. Moreover, healthcare providers can tailor interventions to bolster resilience among mothers, particularly those identified as at risk of developing severe mental health challenges during their pregnancy and post-natal periods. Early identification of mothers who may struggle with developing resilience can lead to timely and targeted interventions, potentially mitigating mental health challenges.

The insights from this analysis can inform health policy, advocating for the inclusion of resilience-building programs in maternal healthcare policies, especially in LMICs where resources are scarce. Health programs can be designed to incorporate resilience training and support as core components, ensuring that maternal mental health is prioritized and addressed comprehensively. Educating healthcare professionals about the concept of maternal resilience and its defining attributes can enhance their ability to support and promote resilience among mothers.

Future research should consider longitudinal studies on maternal resilience, exploring cultural contexts and broader determinants like socio-economic factors and healthcare access. Addressing the high rates of maternal suicide in LMICs by adapting resilience strategies to these settings is vital. Integrating these findings into practice, research, and policy can better support first-time mothers, improving maternal and child health globally.

Discussion and Conclusion

  1. The discussion and conclusion seem a bit wordy and could use editing to make it more concise, especially the implications section.

We tried to make recommended changes in the discussion and implications and reduced the word count.

The concept of maternal resilience for mothers is multifaceted and complex. Based on the concept analysis, we tend to define maternal resilience as the capacity of mothers to persevere with a sense of optimism, striving for a fulfilling existence while nurturing their newborn.

Results of this paper exhibited that maternal resilience in the LMICs context is influenced by multiple factors, including hardships and troubles during pregnancy and the post-natal period, unaddressed health necessities, lack of access to resources, past traumatic events related to childbirth, helplessness, social isolation/ abandonment [49, 52]. Coping with these factors effectively requires some specific characteristics, including social adaptation, support system optimism and mindfulness, which takes the women toward confidence in themselves, motivation towards life and overall health improvements [40,41].

The accessibility of adequate healthcare services for pregnant women in low socioeconomic areas emerges as a critical factor in mitigating stress and promoting maternal resilience. The findings of this study underscore the role of familial support, peer networks, and the availability of responsive healthcare providers as pivotal elements in assisting these women to thrive amidst adversity. However, the data reveal a concerning trend: a subset of these women lacks such support systems, which exacerbates mental health issues within their already stressed psyches [3,5,6,7,10,29,51]. This absence of support undermines their confidence and positivity in life [26,29,45]. In contrast, a robust support network has been consistently identified in the literature as instrumental in fostering maternal well-being [2,13].

Women who experience neonatal death are particularly susceptible to postpartum depression, necessitating more comprehensive treatment approaches to facilitate their return to a stable life [2,9,19]. Family and spousal support are significantly impactful in managing such traumatic experiences effectively. Moreover, low birth weight has been identified as a significant marker of a woman's vulnerability to mental health issues, suggesting that interventions from healthcare providers and government initiatives to ensure proper nutrition for pregnant women in LMICs could enhance resilience to these stressors [9,19.40,43,49,52]. Assurance regarding the health of their baby enables women to build resilience against other adversities [3,6.9].

The role of the woman's immediate social environment, including in-laws and spouses, is also of paramount importance [42,47,48,53]. Education and counselling for these family members about the hormonal changes and challenges faced by pregnant women can foster an environment of empathy and support [40,51]. Such understanding could encourage more practical assistance with household tasks and provide the moral support that is often lacking in many South-Asian LMICs [5,6,7,9,48,54].

Additionally, hospitals and community centers in LMICs frequently face issues of overcrowding and understaffing, leading to healthcare providers experiencing burnout [5,9,43]. This, in turn, can result in a diminished capacity for empathetic and attentive patient care. It is imperative that policies be implemented to manage the high patient-to-staff ratio effectively [28,34,54,55].

Finally, the development of maternal resilience is not solely dependent on the woman; it is inextricably linked to a multitude of internal and external antecedents [14,18,39]. Therefore, comprehensive interventions must address all these dimensions, which is only feasible through collaborative efforts encompassing the pregnant woman, her environmental context, and the healthcare infrastructure of LMICs [19,34,56].

Enhancing adaptation and resilience involves conducting assessments to determine vulnerability, risk, and protective factors. Nurses and other healthcare providers can assist mothers in assessing their personal conditions, cultural beliefs and attitudes, and community and societal conditions [25,39,40,43,57]. Once assessed, providers could work collaboratively to assist mothers in developing a plan for reducing risk and increasing protective factors [34,44,58].

4.1. Implications

Understanding the factors that contribute to resilience enables the development of personalized healthcare plans. Moreover, healthcare providers can tailor interventions to bolster resilience among mothers, particularly those identified as at risk of developing severe mental health challenges during their pregnancy and post-natal periods. Early identification of mothers who may struggle with developing resilience can lead to timely and targeted interventions, potentially mitigating mental health challenges.

The insights from this analysis can inform health policy, advocating for the inclusion of resilience-building programs in maternal healthcare policies, especially in LMICs where resources are scarce. Health programs can be designed to incorporate resilience training and support as core components, ensuring that maternal mental health is prioritized and addressed comprehensively. Educating healthcare professionals about the concept of maternal resilience and its defining attributes can enhance their ability to support and promote resilience among mothers.

Future research should consider longitudinal studies on maternal resilience, exploring cultural contexts and broader determinants like socio-economic factors and healthcare access. Addressing the high rates of maternal suicide in LMICs by adapting resilience strategies to these settings is vital. Integrating these findings into practice, research, and policy can better support first-time mothers, improving maternal and child health globally.

Reviewer 2 Report

Comments and Suggestions for Authors

Dear Authors:

Thank you for allowing me to review this manuscript. I think the manuscript is interesting, but it needs some changes. I am enclosing some comments with the aim of improving this manuscript.

Abstract:

The acronym LMICs is not explained. It is not common to cite authors in the abstract (line 23- line 24).

Introduction:

The introduction is adequate, although perhaps the bibliographic support could be improved. Several citations are more than 5 years old. If possible, improve it. The first sentence, where specific data on mental health problems in this population is provided, is not supported by a reference (line 32-33).

Materials and Methods

This is not a systematic review, but the data collection section could be improved. Search dates could be provided, specifying whether the terms used were mesh terms or free terms, as well as additional information on how many researchers performed the reading and selection of records. In other studies using this Concept Analysis approach, a flow chart is usually provided. When you refer to a manual search (line 120), do you mean a snowball search from a few studies identified in the initial search? If so, this should be better explained.

-In studies with a qualitative component, it is important to have some information about the research team (reflexivity domain). Information such as the number of researchers, experience in this field, etc. should be provided (even if only briefly), and each step should be better explained in relation to the topic studied. I am sending you the reference of a paper that can serve as an example (see data analysis):

Tara Leigh Moore. Resilience of individuals with chronic illness who reside in low resource communities: a concept analysis. International Journal of Nursing Studies Advances,Volume 7, 2024, https://doi.org/10.1016/j.ijnsa.2024.100215.

(https://www.sciencedirect.com/science/article/pii/S2666142X24000420

Results

Together with the flow chart, a table with a description of the included studies, at least indicating the country, the study design and some general characteristics of each study would also help to improve the understanding of the work and to be able to assess from which sources the concepts arise. A narrative description finally explaining how many studies or documents were finally used for the study would also be useful.

Table 1 should contain the references of the documents from which these characteristics have been established. The previous article can also serve as an example in this regard (see table 2).

I think the manuscript is interesting, but there is considerable room for improvement, especially in the presentation of the results. I hope these comments will be helpful. Best regards

Author Response

Abstract:

The acronym LMICs is not explained. It is not common to cite authors in the abstract (line 23- line 24).

We have made changes. Initially, we included the year because we were discussing a specific methodology proposed by Walker and Avant. However, we have now removed the year and are only using the researchers' names.

Introduction:

The introduction is adequate, although perhaps the bibliographic support could be improved. Several citations are more than 5 years old. If possible, improve it. The first sentence, where specific data on mental health problems in this population is provided, is not supported by a reference (line 32-33).

We have added more recent literature to our references and added the references with all citations.

Materials and Methods

This is not a systematic review, but the data collection section could be improved. Search dates could be provided, specifying whether the terms used were mesh terms or free terms, as well as additional information on how many researchers performed the reading and selection of records. In other studies that use this concept analysis approach, a flow chart is usually provided. When you refer to a manual search (line 120), do you mean a snowball search from a few studies identified in the initial search? If so, this should be better explained.

-In studies with a qualitative component, it is important to have some information about the research team (reflexivity domain). Information such as the number of researchers, experience in this field, etc. should be provided (even if only briefly), and each step should be better explained in relation to the topic studied. I am sending you the reference of a paper that can serve as an example (see data analysis):

Tara Leigh Moore. Resilience of individuals with chronic illness who reside in low resource communities: a concept analysis. International Journal of Nursing Studies Advances,Volume 7, 2024, https://doi.org/10.1016/j.ijnsa.2024.100215.

(https://www.sciencedirect.com/science/article/pii/S2666142X24000420

Thank you for the detailed and much needed feedback. We have made major changes to our material and methods section and also provided Prisma flow diagram.

A total of 3736 citations were retrieved from various databases, with an additional 13 citations sourced from references, bringing the initial total to 3749. After removing 1717 duplicate references, 2019 studies remained for screening. Title and abstract screening excluded 1739 citations, leaving 280 articles for full-text retrieval. Out of these, 277 studies were assessed for eligibility. Subsequently, 268 studies were excluded, with 267 not related to maternal resilience and 15 not conducted in low- and middle-income countries (LMICs). This process resulted in 7 studies being included for data extraction and analysis. The data extracted from these studies are intended to provide comprehensive insights into maternal resilience during and after pregnancy.

The analysis revealed that the included studies utilized the term 'maternal resilience.' However, none of these studies offered a clear definition of the term in the context of low- and middle-income countries. It seems that the authors employed 'maternal resilience' to link the idea of resilience specifically to the perinatal period and motherhood rather than identifying unique qualities or aspects of resilience relevant to this life stage being located in a low- and middle-income country. This highlights the necessity for a more precise and consistent definition of 'maternal resilience' in future research to adequately reflect its distinctive characteristics during and after pregnancy.

Results

Together with the flow chart, a table with a description of the included studies, at least indicating the country, the study design and some general characteristics of each study, would also help to improve the understanding of the work and to be able to assess from which sources the concepts arise. A narrative description finally explaining how many studies or documents were finally used for the study would also be useful.

Table 1 should contain the references of the documents from which these characteristics have been established. The previous article can also serve as an example in this regard (see table 2).

We have addressed this in the previous section. Additionally, the table is shared in response to the first reviewer's comments.

Authors

Year

Country

Study Design

Titles

Dadi et al

2020

Ethiopia

Community-based cohort study

Effect of antenatal depression on adverse birth outcomes in Gondar town, Ethiopia: A community-based cohort study.

Dadi et al.

2020

Ethiopia

Community-based cohort study

Antenatal depression and its potential causal mechanisms among pregnant mothers in Gondar town: Application of structural equation model.

Shawon et al

2024

Nepal

Quantitative cross sectional

Role of women empowerment on mental health problems and care-seeking behavior among married women in Nepal: secondary analysis of nationally representative data.

Nabwera et al

2018

Gambia

Case-control study 

The influence of maternal psychosocial circumstances and physical environment on the risk of severe wasting in rural Gambian infants: a mixed methods approach.

Bhamani et al.

2023

Pakistan

Multi-phase Interventional Research

Development and Validation of Safe Motherhood-Accessible Resilience Training (SM-ART) Intervention to Improve Perinatal Mental Health

Abera et al.

2023

Ethiopia

Quantitative cross-sectional study

Stress and resilience during pregnancy: A comparative study between pregnant and non-pregnant women in Ethiopia

Bhamani et al

2024

Pakistan

Single-blinded block randomized controlled study

Promoting mental wellbeing in pregnant women living in Pakistan with the Safe

Motherhood—Accessible Resilience Training (SM‑ART) intervention: a randomized controlled trial

Round 2

Reviewer 1 Report

Comments and Suggestions for Authors

The authors have sufficiently responded to comments, however there are some editing and formatting issues left. 

1.    line 1: low should not be capitalized

2.    line 35: "Low- and Middle-Income Countries" first letters should not be capitalized

3.    some paragraphs are indented and some are not. (example – paragraph starting on line 54 in indented and paragraph starting on line 68 is not). Choose one style and make it consistent throughout manuscript.

4.    There are also spaces between some paragraphs and not between others.

5.    Table 1 format should be similar to table 2 with lines. Also make sure font, size, and bold is same for column headings.  

6.    Line 200: references at end need a comma between them.

7.    Line 204: extra space after researchers

8.    Line 264: extra space after researchers

9.    Line 276: extra space after colleagues

10.  For citations in text, sometimes there is a space between comma and number and sometimes there is not. It should be consistent throughout.

11. Line 386 and 387 citations have ‘.’ Instead of comma.

12. Line 434: fix citation end bracket

Comments on the Quality of English Language

Author Response

  1. line 1: low should not be capitalized
  2. line 35: "Low- and Middle-Income Countries" first letters should not be capitalized
  3. some paragraphs are indented, and some are not. (example – paragraph starting on line 54 in indented and paragraph starting on line 68 is not). Choose one style and make it consistent throughout manuscript.
  4. There are also spaces between some paragraphs and not between others.
  5. Table 1 format should be similar to table 2 with lines. Also make sure font, size, and bold is same for column headings.  
  6. Line 200: references at end need a comma between them.
  7. Line 204: extra space after researchers
  8. Line 264: extra space after researchers
  9. Line 276: extra space after colleagues
  10. For citations in text, sometimes there is a space between comma and number and sometimes there is not. It should be consistent throughout.
  11. Line 386 and 387 citations have ‘.’ Instead of comma.
  12. Line 434: fix citation end bracket

Thank you very much. We really appreciate your criticality in providing feedback. We have corrected all the suggested errors in our manuscript.

Reviewer 2 Report

Comments and Suggestions for Authors

Dear Authors :

Thank you for allowing me to review your manuscript again. I believe that most of the issues raised in the first review have been answered and for that I congratulate you. However, I would like to make a couple of additional suggestions if you would like to take them into consideration:

-Do not use the acronym in the title. I suggest as the final title: A concept Analysis of maternal resilience against pregnancy related mental health challenges in Low- and middle-income countries

-In the method section, when explaining the data analysis, you could include which researchers carried out the analysis and provide information about their experience in this type of research. This research has an important qualitative component and I still think that a minimum of information should be provided in this respect.

-Table 1 can be improved and provides very little information. In table 1 the author and year column could be unified. The column where the title is given makes little meaning: it would be better to add another column where information on the study population is provided (e.g. sample size, characteristics of this population) and another column where the objectives of each study are given, for example.

-Errata in the word resilience in lines 192-193.

Best regards.

Author Response

Do not use the acronym in the title. I suggest as the final title: A concept Analysis of maternal resilience against pregnancy related mental health challenges in Low- and middle-income countries.

Thank you very much, we have made the suggested change. Now the title is:

A concept Analysis of maternal resilience against pregnancy-related mental health challenges in Low- and middle-income countries

-In the method section, when explaining the data analysis, you could include which researchers carried out the analysis and provide information about their experience in this type of research. This research has an important qualitative component and I still think that a minimum of information should be provided in this respect.

Thank you. We have added these details:

The authors involved in the data analysis process possess substantial expertise, ensuring a rigorous approach to the study. One author has a robust background in qualitative inquiry and has published numerous qualitative research papers. Another author specializes in maternal and child health and is currently pursuing a Ph.D. with a research focus on maternal health. The third author, who supervised the entire process, is a senior researcher with extensive experience in this area and has published several qualitative and quantitative research papers. Their combined expertise contributed to a meticulous and thorough analysis of the data.

-Table 1 can be improved and provides very little information. In table 1 the author and year column could be unified. The column where the title is given makes little meaning: it would be better to add another column where information on the study population is provided (e.g. sample size, characteristics of this population) and another column where the objectives of each study are given, for example.

Thank you, we have added this detail now:

Authors & Year

Country

Study Design

Study Objectives

Population and sample

Titles

Dadi et al, 2020

Ethiopia

Community-based cohort study

To identify the incidence and predictors of adverse birth outcomes, focusing on the outcomes resulting from exposure to antenatal depression and other psychosocial risk factors during pregnancy in Gondar Town, Ethiopia

916 pregnant women

Effect of antenatal depression on adverse birth outcomes in Gondar town, Ethiopia: A community-based cohort study.

Dadi et al, 2020

Ethiopia

Community-based cohort study

Aim to explore the causal mechanisms underlying antenatal depression in Gondar, Ethiopia.

916 pregnant women

Antenatal depression and its potential causal mechanisms among pregnant mothers in Gondar town: Application of structural equation model.

Shawon et al, 2024

Nepal

Quantitative cross-sectional

This study investigates the associations between women empowerment and the prevalence of mental health symptoms and care-seeking behavior among ever-married Nepalese women aged 15–49 years.

5556 married women

Role of women empowerment on mental health problems and care-seeking behavior among married women in Nepal: secondary analysis of nationally representative data.

Nabwera et al, 2018

Gambia

Case-control study 

To explore the influence of maternal psychosocial circumstances and physical environment on the risk of severe wasting in rural Gambia

97 cases and 291 controls

The influence of maternal psychosocial circumstances and physical environment on the risk of severe wasting in rural Gambian infants: a mixed methods approach.

Bhamani et al, 2023

Pakistan

Multi-phase Interventional Research

To develop and validate the Safe Motherhood-Accessible Resilience Training (SM-ART) intervention for pregnant women in Pakistan

17 pregnant women and 8 key informants, including psychologists, psychiatrists, and nurses

Development and Validation of Safe Motherhood-Accessible Resilience Training (SM-ART) Intervention to Improve Perinatal Mental Health

Abera et al, 2023

Ethiopia

Quantitative cross-sectional study

To investigate whether pregnancy is associated with greater stress and lower psychological resilience among women living in Jimma, Southwest Ethiopia.

66 pregnant and 154 non-pregnant women

Stress and resilience during pregnancy: A comparative study between pregnant and non-pregnant women in Ethiopia

Bhamani et al, 2024

Pakistan

Single-blinded block randomized controlled study

To examine the effect of the ‘Safe Motherhood—Accessible Resilience Training (SM-ART)’ on resilience, marital adjustment, depression, and pregnancy-related anxiety

200 pregnant women

Promoting mental wellbeing in pregnant women living in Pakistan with the Safe

Motherhood—Accessible Resilience Training (SM‑ART) intervention: a randomized controlled trial

-Errata in the word resilience in lines 192-193.

We have corrected the error.
